# Two Cases of Temporomandibular Synovial Chondromatosis Associated with *Gli1* Gene Mutation

**DOI:** 10.3390/ijerph19084702

**Published:** 2022-04-13

**Authors:** Taeko Fukutani, Shigeaki Toratani, Taku Kanda, Kensaku Matsui, Sachiko Yamasaki, Kensaku Sumi, Ikuko Ogawa, Souichi Yanamoto

**Affiliations:** 1Department of Oral Oncology, Graduate School of Biomedical and Health Sciences, Hiroshima University, Hiroshima 734-8553, Japan; tora@hiroshima-u.ac.jp (S.T.); sayamasaki@hiroshima-u.ac.jp (S.Y.); ksumi@hiroshima-u.ac.jp (K.S.); syana@hiroshima-u.ac.jp (S.Y.); 2Department of Oral and Maxillofacial Surgery, Hiroshima Prefectural Hospital, Hiroshima 734-8530, Japan; kantaku1209@gmail.com (T.K.); k-matsui@hph.pref.hiroshima.jp (K.M.); 3Center of Oral Clinical Examination, Hiroshima University Hospital, Hiroshima 734-8553, Japan; dlabo@hiroshima-u.ac.jp

**Keywords:** bone tumor of mandibular condyle, cone-beam computed tomography, magnetic resonance spectroscopy, synovial chondromatosis, temporomandibular joint, trismus, zinc finger protein GLI1

## Abstract

Synovial chondromatosis (SC) is a rare benign disease involving multifocal generation of ectopic cartilage in the synovial tissue. Herein, we report two cases of SC in the temporomandibular joint: a 38-year-old woman (patient 1) and 39-year-old woman (patient 2). Both patients had trismus, jaw joint noises, and jaw-opening pain in the temporomandibular joint. Cone-beam computed tomography (CT) and magnetic resonance imaging (MRI) in patient 1 showed multiple calcified loose bodies around the right mandibular condyle. In addition, CT and MRI in patient 2 showed multiple calcified loose bodies around the left mandibular condyle and temporal bone perforation. Following establishing a diagnosis of SC, both patients underwent tumor resection via open surgery. In immunohistochemical examinations of the resected tissues, tumor cells showed intense nuclear staining with labeled anti-*Gli1* antibody. Gene sequencing revealed that both patients had a homozygous mutation in the *Gli1* gene (rs2228226 G>C). In conclusion, we suggest that the *Gli1* gene (rs2228226 G>C) may be involved in the etiology of SC.

## 1. Introduction

Synovial chondromatosis (SC) is a benign neoplastic disease that involves ectopic cartilage generation and exhibits multifocal lesions in the synovial tissue. This condition affects the knee joint in 50% of cases. However, SC is relatively rare in the temporo-mandibular joint (TMJ) [1,2]. SC consists of a synovial metaplasia, which affects 1 in 100,000 people [3]. In large joints, such as the knee, hip, shoulder, and elbow, men are four times more likely than women to experience SC [4]. However, SC of the TMJ occur primarily in women [5,6].

The main clinical manifestations include restricted mandibular motion, pain, swelling of the anterior surface of the auricle, and crepitus. Notably, these symptoms are nonspecific, which is why delayed diagnosis and misdiagnosis are common. Carls et al. reported that 6 of 1785 patients diagnosed with TMJ had this disorder, and the similarity of the symptoms of this disorder to those of TMJ disorders (TMD)s also contributes to the difficulty in diagnosis [7]. The primary treatment for SC is surgery. Approximately 240 cases of TMJ SC have been reported in the English language literature, but its etiology has not been elucidated [5]. Although SC was classified as a joint lesion in the category of bone tumors in the 2002 World Health Organization (WHO) classification of tumors, it was recognized as a chondrogenic tumor in the 2013 classification [8,9]. One of the reasons is that SC has been reported to be an abnormality of the hedgehog (Hh) signaling system. However, there are few reports on genetic approaches for SC.

We believe that by clarifying the etiology of the disease, we can contribute to the diagnosis and management of patients affected by this disease. Herein, we describe two cases of SC of the TMJ with a focus on immunohistochemical examination and gene-related considerations of patched 1 and Gli1 and discuss the relevant literature.

## 2. Case Reports

### 2.1. Patient 1

A 38-year-old woman was referred to our hospital for a complaint of joint noise and jaw-opening pain for 11 years in the right TMJ. At the first visit, in September 2013, she had crepitus and exhibited mandibular shift to the right while opening her mouth. The mouth-opening capacity was 20 mm. Cone-beam CT (CBCT) examination revealed bone resorption at the right head of the mandible and glenoid fossa and a large amount of granular radiopaque material around the right head of the mandible (Figure 1a). The patient or her family had no relevant medical history. T2-weighted magnetic resonance imaging (MRI) showed granular cell tumor with high signal intensity in the right TMJ (Figure 1b). Therefore, we clinically diagnosed the patient with SC of the right TMJ S/O.

In November 2013, we performed a preauricular incision and tumor resection using open joint capsule surgery under general anesthesia. When we opened the superior joint space, a pale-yellow clear transparent viscous liquid flowed out, together with 1–3 mm grayish-white spherical hard solid particulates resembling rice bodies. We removed the tumor and hard solids from the inside of the right mandibular head and anteroposterior region, including the mandibular head, synovial membrane, and joint disc. Gross pathology of the excised specimens indicated that the tumor extended to the synovial membrane, and a hard solid mass was palpable in the tissue (Figure 2).

From three weeks after intermaxillary fixation, the patient maintained an occlusal relationship with a splint and actively performed mouth-opening training. Although she exhibited a mandibular shift to the right while opening her mouth, her mouth-opening capacity improved to 36 mm. No recurrence was observed 8 years after surgery, and the patient had a favorable prognosis (Figure 3).

Histopathological examination of formalin-fixed paraffin-embedded tissue sections stained with hematoxylin–eosin revealed fibrous connective tissues that were identified as synovial membranes, with hyaline cartilage nodules inside. The synovium covering the surface of the mandibular head showed similar deformities, and this surface exhibited bone resorption. Although the chondrocytes showed a diverse range of nuclear sizes, they did not contain mitotic figures, and there were no malignant findings. The loose bodies in the joint space were hyaline cartilage and endochondral ossification. Therefore, a histopathological diagnosis of SC was established.

On immunohistochemical staining, tumor cell nuclei showed intense staining with labeled anti-Gli1 antibody (Figure 4). 

Immunostaining using labeled anti-PTCH1 antibody revealed strong staining in the whole tissue, with no specific findings. We used rhGLI-1 (diluted 1/100: aa 1-234, R&D Systems, Minneapolis, MN, USA), anti-PTCH1 (diluted 1/50: ab53715: rabbit polyclonal, Abcam, Cambridge, UK), and anti-PTCH1 (diluted 1/50: sc6149: mouse polyclonal antibody, Santa Cruz Biotechnology, Dallas, TX, USA). 

We extracted DNA from the excised specimens and blood from the patient and analyzed the sequence of the entire *Gli1* gene. We used QIAmp blood mini kit (QIAGEN N.V., Hulsterweg, Venlo, The Netherlands) for analyzing blood and QIA DNA mini kit (QIAGEN) for excised tissues. The *Gli1* gene comprised 12 exons, as demonstrated through the Sanger method of DNA sequencing. Polymerase chain reaction amplification was performed using DNA as templates. Forty cycles were performed, with one cycle as 98 °C for 10 s (denaturation), 60 °C for 30 s (annealing), and 65 °C for 30 s (extension).

In addition, we used 5′-GGGGCAAATAGGGCTTCACA-3′ (forward primer sequence) and 5′-GTGAGAGGCTGGTGTTCTGT-3′ (reverse primer sequence) to assess exon 12 of *Gli1* gene.

Interestingly, both the excised specimens and blood from the patient showed a point mutation from G to C in exon 12 of *Gli1* (rs2228226 G>C), encoding a change from glutamic acid at amino acid position 1100 to glutamine (E1100Q). Furthermore, the patient had C/C genotype, reflecting a homozygous mutation (Figure 5a,b).

### 2.2. Patient 2

A 39-year-old woman was referred to our hospital for a 5-year complaint of joint noise and jaw-opening pain in the left TMJ. At the first visit, in May 2015, she had crepitus and exhibited mandibular shift to the left while opening her mouth. Her mouth-opening capacity was 26 mm. Neither she nor her family had relevant medical history.

A tumor in the joint cavity was detected on CT. In addition, part of the temporal bone exhibited bone resorption, and the inner surface of the skull was perforated (Figure 6a).

A similar tumor was observed on MRI examination, and the boundary was clear, except for the glenoid fossa. Small amounts of calcification and fluid retention were observed in the lesions. Although the tumor was in contact with the dura mater in the upper part, there was no progression to the brain parenchyma (Figure 6b).

Therefore, a clinical diagnosis of SC of the left TMJ S/O was made.

In August 2015, we performed preauricular incision and tumor resection, including the synovial disc and left head of the mandible, using open joint capsule surgery under general anesthesia. When we incised the upper joint space, we observed a cartilage-like hard solid mass. Gross pathology during excision revealed that the mandibular shape was relatively maintained. Loose bodies were palpable in the synovial tissue. Moreover, several loose bodies were observed in the superior joint space (Figure 7).

Bone resorption was observed 7 mm above the glenoid fossa. Since part of the dura mater was exposed, we evaluated this by consulting a brain surgeon. However, no perforation of the dura mater or spinal fluid leakage was observed.

From three weeks after intermaxillary fixation, the patient maintained an occlusal relationship with a splint and actively performed mouth-opening training. Although she exhibited mandibular shift to the left while opening her mouth, her mouth-opening capacity improved to 35 mm. No recurrence was observed 6 years after surgery, and the patient had a favorable prognosis (Figure 8).

Histopathological findings, immunohistochemical staining findings, and *Gli1* gene sequence analysis (Figure 9 and Figure 10a,b) were the same as those in patient 1.

## 3. Discussion

SC of the TMJ is often diagnosed late because of the absence of physical signs and symptoms. It may also be misdiagnosed because its symptoms are similar to those of TMJ disorders. The mean incubation period is long, with TMJ SC having an average onset time of more than 1 year [10]; some studies have reported onset times of about 20 years [11,12]. Similarly, in our cases, it took 11 years for patient 1 and 5 years for patient 2 to be diagnosed. Conventional radiographs, the first-line method of diagnosis of TMD, may not detect cartilaginous nodules lacking calcification or ossification. In cases of refractory TMD, SC must be suspected and CBCT, CT, and MRI must be performed for diagnosis.

The treatment of SC is mainly surgical. Surgery can be broadly classified into the following categories: (1) tumor removal only (including curettage on the synovial surface), (2) tumor removal + synovectomy, (3) tumor removal + synovectomy + discectomy, (4) tumor removal + synovectomy + mandibulectomy, and (5) tumor removal + synovectomy + disc + mandibulectomy. The location of the lesion in the joint cavity, size of the tumor, continuity with the synovial membrane, presence of disc perforation or mandibular deformity, and postoperative esthetics and jaw function are all taken into consideration in selecting the surgical technique. The reason why only tumor excision is performed in some cases is that this condition has a low recurrence rate [10]. On the other hand, the reason for synovectomy is the continuity of the lesion with the synovium or tumor extension. Recurrence of this disease has been attributed to incomplete excision of the lesion or leftover free bodies [13]. Ballard et al. stated that complete excision of the tumor or free body and synovectomy are difficult in the narrow TMJ [14].

In both of our cases, tumors developed extensively, and perforation of the disc was suspected; thus, we opted for tumor removal + synovial + disc + mandibular capsulotomy. Currently, both patients show no signs of recurrence and have no postoperative functional impairment.

SC was reclassified from the category of joint lesions into the category of chondrogenic tumors of bone in the 2013 WHO classification for several reasons. These reasons are that SC shows, in order of frequency, clonal karyotypic abnormality in chromosome 6, rearrangements of 1p22 and 1p13, and extra copies of chromosome 5 [15]. Furthermore, disruption of the Hh signaling pathway is a feature of benign cartilage tumors. In addition, abnormality in the Hh signaling pathway has been reported in patients with SCs [9]. Recently, it was reported that the fibronectin 1 and activin receptor 2A fusion gene identified in chondrosarcoma were also identified in SC [16,17,18]. However, there are few reports of genetic approaches for SC.

Hopyan et al. [19] observed that mice lacking the suppressor gene *Gli3* of the Hh signaling pathway developed SC. They also compared the expression levels of Hh receptor *PTCH1* and downstream signal molecule *Gli1* in human diseases with those in healthy individuals and found that the expression of *PTCH1* was increased by eight-fold and *Gli1* by six-fold. Mutation in the *PTCH1* or *Gli1* gene is considered a cause of Hh pathway disruption. However, no such mutations have been identified to date.

The rs2228226 G>C point mutation in the *Gli1* gene is a non-synonymous single nucleotide polymorphism near the transactivation region on the C-terminal side of the protein. Since glutamic acid at amino acid position 1100 (1100E) provides a negative charge in the side chain, its substitution with glutamine (1100Q) causes a significant change in the charge, thereby affecting the transactivation region [20]. According to a study by Lees et al. [21], glutamine (1100Q) increased transcription activity by approximately 50% compared with glutamic acid (1100E), which is consistent with the finding that the Hh pathway was activated in this case.

Furthermore, Szkandera et al. [22] reported that patients with C/C genotype of *Gli1* (rs2228226 G>C) had greater short-term recurrence than those with G/G and G/C genotypes in a cohort study of colon cancer recurrence. Li et al. [23] found that patients with lymphocytic leukemia who had the C/C genotype of *Gli1* (rs2228226 G>C) were 3.3 times more likely to develop lymphocytic leukemia than were healthy individuals. Both of the present cases had C/C genotype, suggesting an association between SC and C/C genotype of *Gli1* (rs2228226 G>C).

## 4. Conclusions

SC of the TMJ is a rare disease; the nonspecific nature of its symptoms makes diagnosis difficult, detection is often delayed, and patients suffer for a long period of time.

With the aim of helping to solve this problem, we performed a genetic analysis. Immunohistochemical staining of tumor cells showed intense nuclear staining with labeled anti-*Gli1* antibody in the two patients. Furthermore, in *Gli1* gene analysis, both patients had a homozygous mutation (C/C genotype) in *Gli1* (rs2228226 G>C). Therefore, an association between SC and the C/C genotype of *Gli1* (rs2228226 G>C) might be suggested.

Our results suggest the usefulness of a genetic approach for SC. Since SC of the TMJ is rare, the results suggest that further research is needed, including the creation of a multicenter study model.

## Figures and Tables

**Figure 1 ijerph-19-04702-f001:**
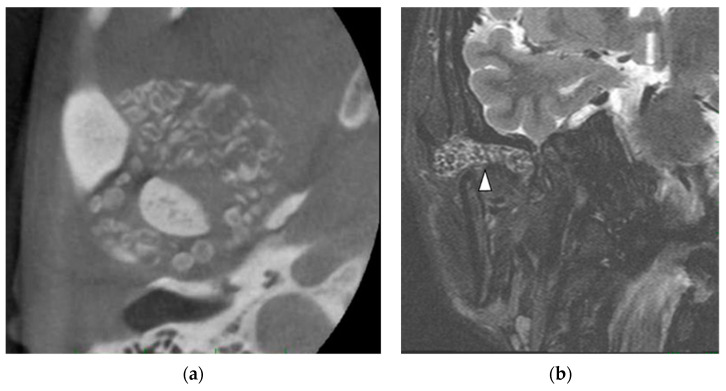
Imaging results for patient 1. (**a**) Cone-beam computed tomography. Bone resorption is observed at the right head of the mandible and glenoid fossa, together with a large amount of granular radiopaque material around the right head of the mandible. Arrows indicate lesions. (**b**) Magnetic resonance imaging. A granular tumorous lesion of approximately 30 mm with high signal intensity is visible at the right temporomandibular joint in the T2-weighted image. Arrows indicate lesions.

**Figure 2 ijerph-19-04702-f002:**
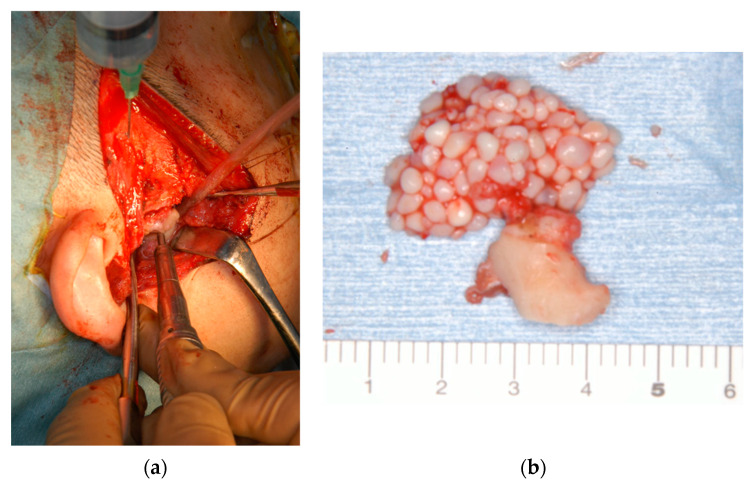
Photographs of intraoperative and excised material in patient 1. (**a**) Intraoperative photograph. We performed a preauricular incision and tumor resection, including the synovial disc and right head of the mandible by open-joint capsule surgery under general anesthesia. (**b**) Excised material from patient 1. The mandibular head was deformed, and the disc was perforated. A total of 151 free hard solid objects resembling rice bodies were found in the superior joint space.

**Figure 3 ijerph-19-04702-f003:**
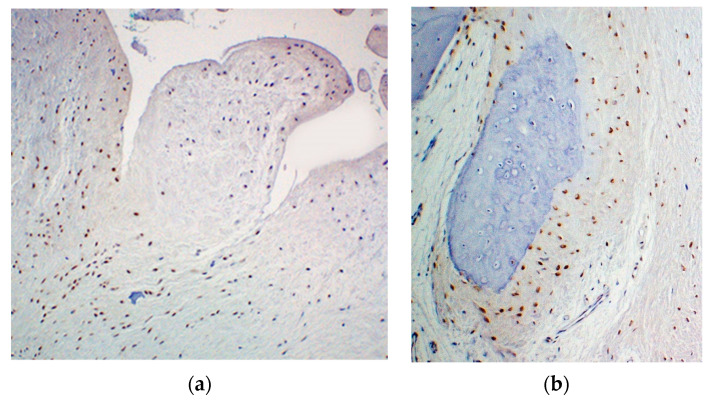
Immunohistochemical findings in patient 1. Gli1 was expressed in the nuclei of the neoplastic cells (**a**), especially at the periphery of the cartilage nodule (**b**).

**Figure 4 ijerph-19-04702-f004:**
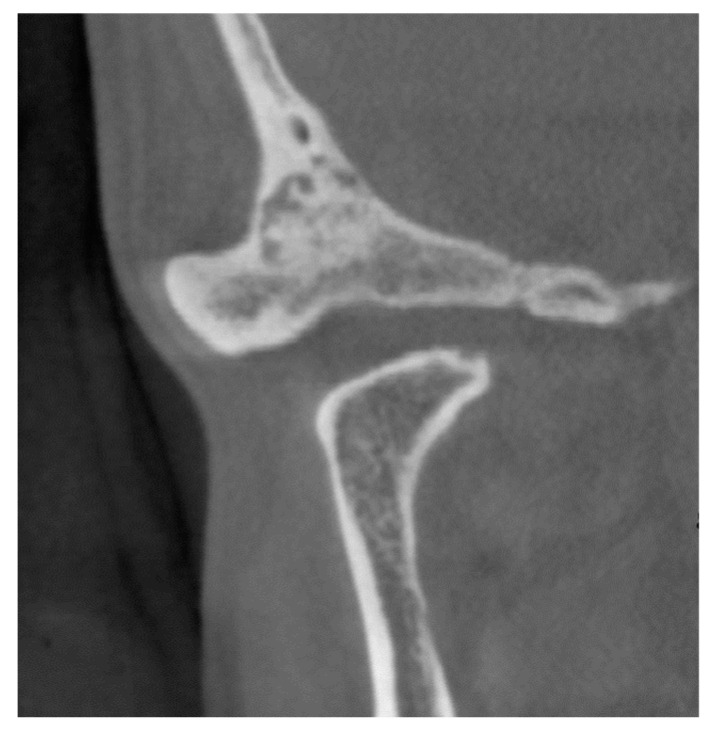
Imaging results in patient 1 after surgery. Cone-beam computed tomography. Eight years after surgery, there is no recurrence.

**Figure 5 ijerph-19-04702-f005:**
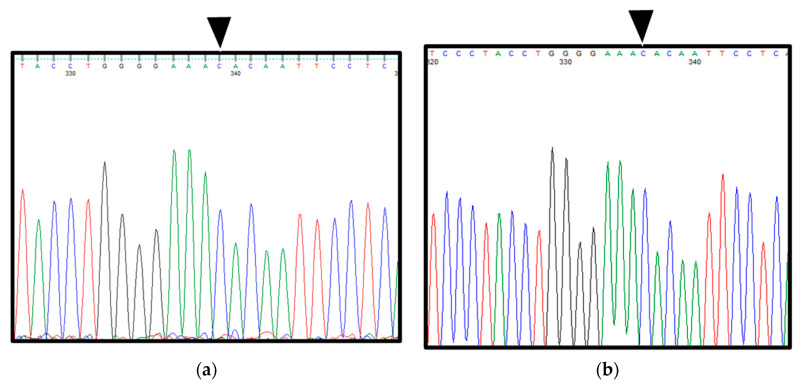
Analysis of the sequence of the entire *Gli1* gene in the patient’s excised samples and blood based on Sanger sequencing. (**a**) DNA sequencing of blood in patient 1. (**b**) DNA sequencing in excised specimens in patient 1 (synovial tissue). Black arrows indicate a point mutation site from G to C in exon 12 of *Gli1*.

**Figure 6 ijerph-19-04702-f006:**
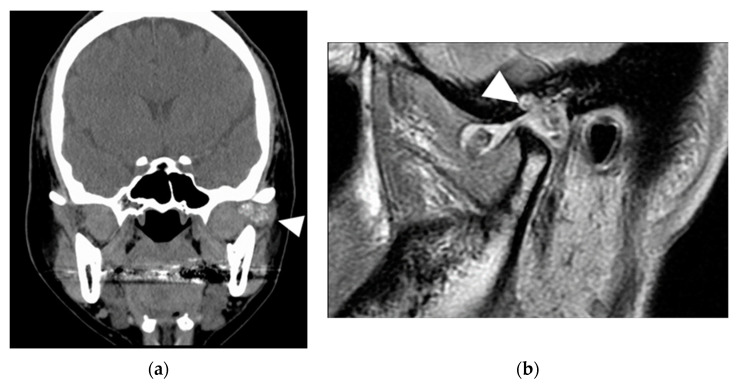
Imaging results in patient 2. (**a**) Computed tomography. Anterior and posterior deflection of the joint head is observed, with temporal bone resorption and perforation of the cranium. (**b**) Magnetic resonance imaging. A tumor similar to that detected on computed tomography is observed. Although it is in contact with the dura in the upper part, no progression to the brain parenchyma is seen. White arrows indicate lesions.

**Figure 7 ijerph-19-04702-f007:**
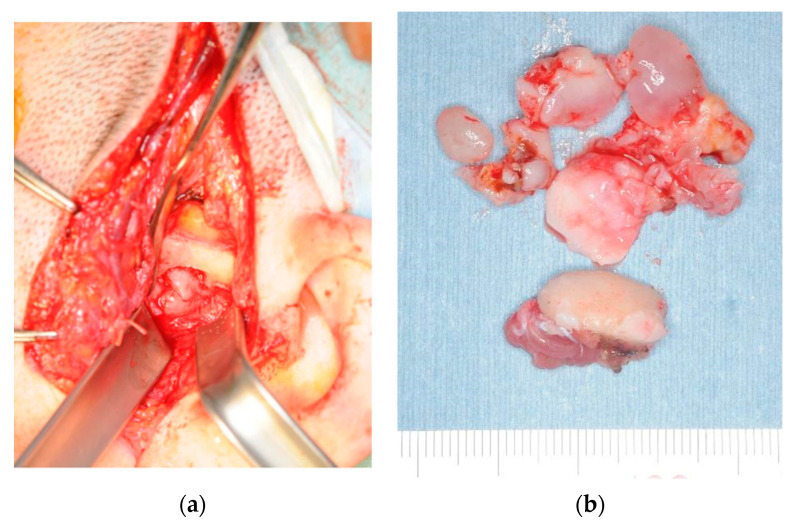
Photographs of intraoperative and excised material in patient 2. (**a**) Intraoperative photograph showing preauricular incision and tumor resection, including the synovial disc and left head of the mandible, using open-joint capsule surgery under general anesthesia. (**b**) Excised tissues from patient 2. Several hard masses, 10–20 mm in diameter, are observed in the superior joint space.

**Figure 8 ijerph-19-04702-f008:**
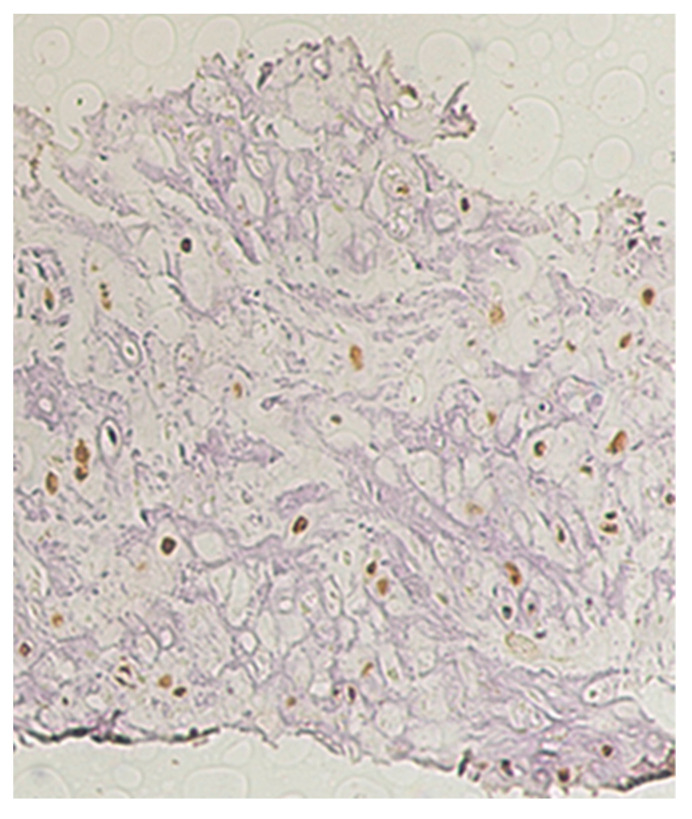
Immunostaining findings in patient 2. Immunostaining with labeled anti-Gli1 antibody. These findings are similar to those in patient 1.

**Figure 9 ijerph-19-04702-f009:**
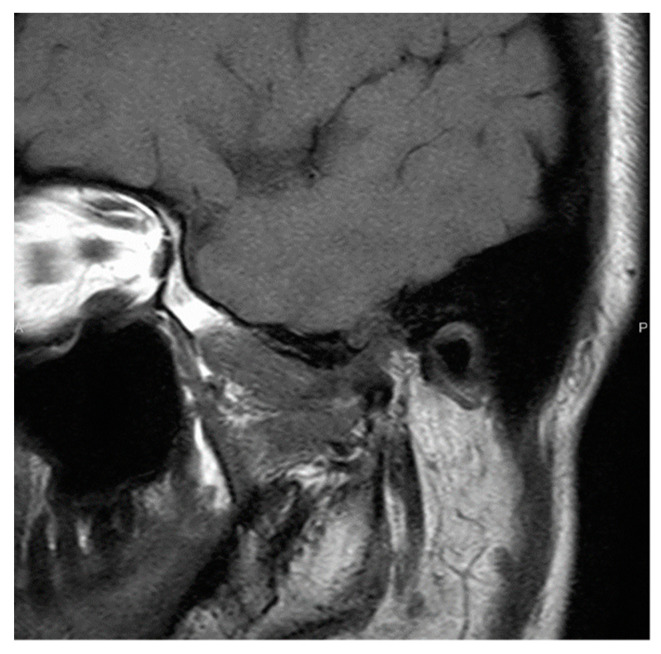
Imaging results in patient 2 after surgery. Magnetic resonance imaging. Six years after surgery, there is no recurrence.

**Figure 10 ijerph-19-04702-f010:**
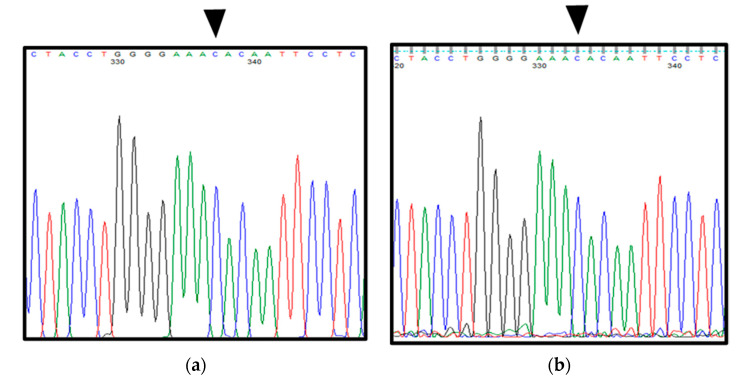
Analysis of the sequence of the entire *Gli1* gene in the excised samples and blood in patient 2 based on the Sanger sequencing. (**a**) DNA sequencing in blood in patient 2. (**b**) DNA sequencing in excised samples in patient 2 (synovial tissue). Of note, both the excised material and blood show a point mutation from G to C at exon 12 (rs2228226 G>C) of *Gli1* with a change from glutamic acid to glutamine (E1100Q) at amino acid position 1100. In addition, the patient has C/C genotype, reflecting a homozygous variant. Black arrows indicate a point mutation site from G to C in exon 12 of *Gli1*.

## Data Availability

Not applicable.

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
