# Peer review of "Two Cases of Temporomandibular Synovial Chondromatosis Associated with Gli1 Gene Mutation"

_ijerph, 2022, doi:10.3390/ijerph19084702_

Round 1

Reviewer 1 Report

  1. Introduction. Please introduce why authors focused on immunohistochemical examination and gene-related considerations of patched 1 and Gli1?
  2. Line 77-80 on Page 2, what is rhG1? two anti-PTCH1?
  3. There are many typing errors, for example, the legend of Figure 4.
  4. Please provide the magnified images of immunostaining. What the cells positive for Gli1 are? 
  5. "Immunostaining using labeled anti-PTCH1 antibody revealed strong staining in the whole tissue, with no specific findings", please explain.
  6. Is Gli1expressed in normal TMJ synovial tissue?

Reviewer 2 Report

In this manuscript, the authors reported  two cases of SC in the temporomandibular joint, and investigated  immunohistochemical examinations of the resected tissues, and gene sequencing of Gli1 gene (rs2228226 G>C) may be involved in the etiology of SC.
The paper is novel in its approach, and  interesting observation.
I believe the paper will be of interest to the readership of  Journal, and would recommend it for acceptance after the minor points listed below.
I hope these comments will be helpful.

1.Introduction
I think you need to clearly articulate why this Gli1 gene  was investigated.
I would anticipate that adding this more complete introduction would also require the addition of a number of additional references.

2. Immunohistochemical examinations
With regard to content, there appears to be a lack of sample number. Please add the sample size of immunohistochemical examination at least 6 or 7 in patients with SC of TMJ.

3.Discussion
The expression of Gli1 gene in the SC of TMJ does not appear to be particularly distinct, and concern only few samples in a limited area.
The authors need to describe the significance of Gli1 gene in SC, especially in the knee joints. It may be appropriate to include references regarding the significance of SC, especially in the knee joints.

Reviewer 3 Report

Generally the paper provides the thorough view on histologic features of the presented cases. The article might be improved providing more information on TMJ SC epidemiology, possible etiology and treatment methods.

Reviewer 4 Report

The manuscript titled "Two rare cases of temporomandibular synovial chondromatosis associated with Gli1 gene mutation" definitely requires thorough revision.

First an foremost, despite it being only a case report, the bibliography still needs to be more extensive as well as more up-to-date. There are only 10 positions in the "References" section and with an exception of one work from 2014, all of them come from the early 2000s and some are even older than that. This needs to by addressed by the authors and corrected.

In the "Introduction" there should be a paragraph added that touches the subject of synovial chondromatosis which not only describes the condition itself but also presents the current state of knowledge about the disease (e.g. briefly mentioning medical papers/articles published to-date, how did they contribute to the overall understanding of the disease, which areas can still be improved,  what is missing from the literature and what type of research can be of benefit to expanding our knowledge). At the same time it should be explained why did the authors decide to conduct their own analysis of 2 patient cases in the context of how it supplements everything that's already been published on the subject and what new information does this manuscript bring to the table.

The article is also missing the criteria under which the patients were chosen for the study and qualified for surgical procedures - especially taking under consideration that in both patients the scope of resection was different.

The "Discussion" section needs extensive modification because it focuses solely on the genetic aspects of chondromatosis and omits its clinical facets which are extremely important for any practicing surgeon. There is also a lack of discussion as far as the criteria for qualifying the patients for surgical procedures go. There is no description of various approaches alongside their pros/cons both for the patient and the operator. It should be at least shortly mentioned what type of surgical access is suitable for different types of lesions and the criteria that should be followed when deciding on the scope of the procedure. It would be highly beneficial to the manuscript if other authors' experiences, methods and results were mentioned along with the long-term outcomes. 

It would be justifiable if the authors explained why the intermaxillary fixation was used and if any post-operative rehabilitating treatment was administered during the patients' recovery process and if yes - what type of treatment.

Due to the casuistic nature of this type of paper it is necessary to include intraoperative photographs as well as a comparison of CT/NMR scans (in at least two different screenings) taken before and after the surgery.

The "Conclusions" need to be completely rewritten because at the current state it is a repetition of all that has been already put in the results and introduction and it doesn't seem to stem from the entirety of the manuscript.

The use of English is generally correct and understandable but it needs more attention and editorial work. There are quite a few repetitions and given how relatively short the manuscript is - these could be easily fixed and avoided. Some terms that were used in the text don't come across as medically appropriate e.g. "hard solid materials" should be replaced with a more scientific phrase. In the Line 115 a mistake was noticed as follows: "There was bone resorption was observed..." and in Figure 4. "Although it in in contact."

As the last suggestion - considering how rare the occurrence of synovial chondromatosis in TMJ is - it would be worth entertaining a possibility of creating a model of a multicenter study as well as presenting the readers with a prognosis of which aspects of this disease should be explored in the future.  

Round 2

Reviewer 1 Report

I have no more comment

Author Response

I would like to thank you for your insightful comments on this paper. Their comments allowed us to significantly improve the paper. Thank you for your continued support.

Reviewer 4 Report

The Authors made  all the suggested modifications and I am satisfied with the final result of the manuscript. My last suggestion is - please avoid the use of personal pronouns such as we, our, etc.

Author Response

I would like to thank you for your insightful comments on this paper. Their comments allowed us to significantly improve the paper. I would like to thank you for your insightful comments on this paper. These comments have allowed us to significantly improve the paper.

In addition, we have engaged Editage to proofread the English text. We attach their certificate for your review. Thank you for your continued support.
